# DropMax: Adaptive Variational Softmax

**Hae Beom Lee**[1,2], **Juho Lee**[3,2], **Saehoon Kim**[2], **Eunho Yang**[1,2], **Sung Ju Hwang**[1,2]
KAIST[1], AItrics[2], South Korea,
University of Oxford[3], United Kingdom,
{haebeom.lee, eunhoy, sjhwang82}@kaist.ac.kr
juho.lee@stats.ox.ac.uk, shkim@aitrics.com

## Abstract

We propose DropMax, a stochastic version of softmax classifier which at each iteration drops non-target classes according to dropout probabilities adaptively decided for each instance. Specifically, we overlay binary masking variables over class output probabilities, which are input-adaptively learned via variational inference. This stochastic regularization has an effect of building an ensemble classifier out of exponentially many classifiers with different decision boundaries. Moreover, the learning of dropout rates for non-target classes on each instance allows the classifier to focus more on classification against the most confusing classes. We validate our model on multiple public datasets for classification, on which it obtains significantly improved accuracy over the regular softmax classifier and other baselines. Further analysis of the learned dropout probabilities shows that our model indeed selects confusing classes more often when it performs classification.

## 1 Introduction

Deep learning models have shown impressive performances on classification tasks [17, 10, 11]. However, most of the efforts thus far have been made on improving the network architecture, while the predominant choice of the final classification function remained to be the basic softmax regression. Relatively less research has been done here, except for few works that propose variants of softmax, such as Sampled Softmax [12], Spherical Softmax [5], and Sparsemax [22]. However, they either do not target accuracy improvement or obtain improved accuracy only on certain limited settings.

In this paper, we propose a novel variant of softmax classifier that achieves improved accuracy over the regular softmax function by leveraging the popular dropout regularization, which we refer to as *DropMax*. At each stochastic gradient descent step in network training, DropMax classifier applies dropout to the exponentiations in the softmax function, such that we consider the true class and a random subset of other classes to learn the classifier. At each training step, this allows the classifier to be learned to solve a distinct subproblem of the given multi-class classification problem, enabling it to focus on discriminative properties of the target class relative to the sampled classes. Finally, when training is over, we can obtain an ensemble of exponentially many [1] classifiers with different decision boundaries.

Moreover, when doing so, we further exploit the intuition that some classes could be more important than others in correct classification of each instance, as they may be confused more with the given instance. For example in Figure 1, the instance of the class *cat* on the left is likely to be more confused with class *lion* because of the lion mane wig it is wearing. The *cat* instance on the right, on the other hand, resembles *Jaguar* due to its spots. Thus, we extend our classifier to *learn* the probability of dropping non-target classes for each input instance, such that the stochastic classifier can consider

classification against confusing classes more often than others adaptively for each input. This helps to better classify such difficult instances, which in turn will results in improving the overall classification performance.

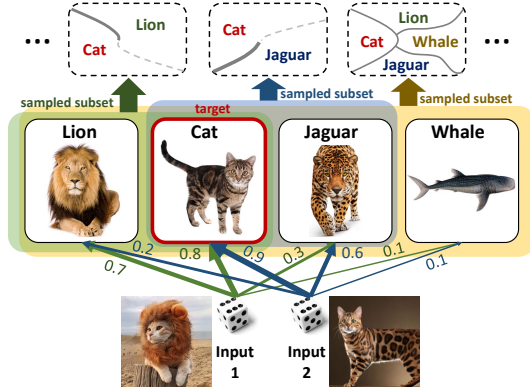

The proposed adaptive class dropout can be also viewed as stochastic attention mechanism, that selects a subset of classes each instance should attend to in order for it to be well discriminated from any of the false classes. It also in some sense has similar effect as boosting, since learning a classifier at each iteration with randomly selected non-target classes can be seen as learning a weak classifier, which is combined into a final strong classifier that solves the complete multi-class classification problem with the weights provided by the class retain probabilities learned for each input. Our regularization is generic and can be applied even to networks on which the regular dropout is ineffective, such as ResNet, to obtain improved performance.

Figure 1: **Concepts.** For a given instance, classes are randomly sampled with the probabilities learned adaptively for each instance. Then the sampled subset participates in the classification.

We validate our model on five public datasets for classification, on which it consistently obtains significant accuracy improvements over the base softmax, with noticeable improvements on fine-grained datasets ($3.38\%p$ on AWA and $7.77\%p$ on CUB dataset.)

Our contribution is threefold:

- We propose a novel stochastic softmax function, DropMax, that randomly drops non-target classes when computing the class probability for each input instance.

- We propose a variational inference framework to adaptively learn the dropout probability of non-target classes for each input, s.t. our stochastic classifier considers non-target classes confused with the true class of each instance more often than others.

- We propose a novel approach to incorporate label information into our conditional variational inference framework, which yields more accurate posterior estimation.

## 2   Related Work

**Subset sampling with softmax classifier**    Several existing work propose to consider only a partial susbset of classes to compute the softmax, as done in our work. The main motivation is on improving the efficiency of the computation, as matrix multiplication for computing class logits is expensive when there are too many classes to consider. For example, the number of classes (or words) often exceeds millions in language translation task. The common practice to tackle this challenge is to use a shortlist of $30K$ to $80K$ the most frequent target words to reduce the inherent scale of the classification problem [3, 20]. Further, to leverage the full vocabulary, [12] propose to calculate the importance of each word with a deterministic function and select top-$K$ among them. On the other hand, [22] suggest a new softmax variant that can generate sparse class probabilities, which has a similar effect to aforementioned models. Our model also works with subset of classes, but the main difference is that our model aims to improve the accuracy of the classifier, rather than improving its computational efficiency.

**Dropout variational inference**    Dropout [25] is one of the most popular and succesful regularizers for deep neural networks. Dropout randomly drops out each neuron with a predefined probability at each iteration of a stochastic gradient descent, to achieve the effect of ensemble learning by combining exponentially many networks learned during training. Dropout can be also understood as a noise injection process [4], which makes the model to be robust to a small perturbation of inputs. Noise injection is also closely related to probabilistic modeling, and [8] has shown that a network trained with dropout can be seen as an approximation to deep Gaussian process. Such Bayesian understanding of dropout allows us to view model training as posterior inference, where

predictive distribution is sampled by dropout at test time [13]. The same process can be applied to convolutional [6] and recurrent networks [7].

**Learning dropout probability**   In regular dropout regularization, dropout rate is a tunable parameter that can be found via cross-validation. However, some recently proposed models allow to learn the dropout probability in the training process. Variational dropout [14] assumes that each individual weight has independent Gaussian distribution with mean and variance, which are trained with reparameterization trick. Due to the central limit theorem, such Gaussian dropout is identical to the binary dropout, with much faster convergence [25, 27]. [23] show that variational dropout that allows infinite variance results in sparsity, whose effect is similar to automatic relevance determination (ARD). All the aforementioned work deals with the usual posterior distribution not dependent on input at test time. On the other hand, adaptive dropout [2] learns input dependent posterior at test time by overlaying binary belief network on hidden layers. Whereas approximate posterior is usually assumed to be decomposed into independent components, adaptive dropout allows us to overcome it by learning correlations between network components in the mean of input dependent posterior. Recently, [9] proposed to train dropout probability for each layer for accurate estimation of model uncertainty, by reparameterizing Bernoulli distribution with continuous relaxation [21].

## 3   Approach

We first introduce the general problem setup. Suppose a dataset $\mathcal{D} = \{(\mathbf{x}_i, \mathbf{y}_i)\}_{i=1}^N$, $\mathbf{x}_i \in \mathbb{R}^d$, and one-hot categorical label $\mathbf{y}_i \in \{0, 1\}^K$, with $K$ the number of classes. We will omit the index $i$ when dealing with a single datapoint. Further suppose $\mathbf{h} = \mathrm{NN}(\mathbf{x}; \omega)$, which is the last feature vector generated from an arbitrary neural network $\mathrm{NN}(\cdot)$ parameterized by $\omega$. Note that $\omega$ is globally optimized w.r.t. the other network components to be introduced later, and we will omit the details for brevity. We then define $K$ dimensional class logits (or scores):

$$\mathbf{o}(\mathbf{x}; \psi) = \mathbf{W}^\top \mathbf{h} + \mathbf{b}, \quad \psi = \{\mathbf{W}, \mathbf{b}\} \tag{1}$$

The original form of the softmax classifier can then be written as:

$$p(\mathbf{y}|\mathbf{x}; \psi) = \frac{\exp(o_t(\mathbf{x}; \psi))}{\sum_k \exp(o_k(\mathbf{x}; \psi))}, \quad \text{where } t \text{ is the target class of } \mathbf{x}. \tag{2}$$

### 3.1   DropMax

As mentioned in the introduction, we propose to randomly drop out classes at training phase, with the motivation of learning an ensemble of exponentially many classifiers in a single training. In (2), one can see that class $k$ is completely excluded from the classification when $\exp(o_k) = 0$, and the gradients are not back-propagated from it. From this observation, we randomly drop $\exp(o_1), \ldots, \exp(o_K)$ based on Bernoulli trials, by introducing a dropout binary mask vector $z_k$ with *retain* probability $\rho_k$, which is one minus the dropout probability for each class $k$:

$$z_k \sim \mathrm{Ber}(z_k; \rho_k), \quad p(\mathbf{y}|\mathbf{x}, \mathbf{z}; \psi) = \frac{(z_t + \varepsilon) \exp(o_t(\mathbf{x}; \psi))}{\sum_k (z_k + \varepsilon) \exp(o_k(\mathbf{x}; \psi))} \tag{3}$$

where sufficiently small $\varepsilon > 0$ (e.g. $10^{-20}$) prevents the whole denominator from vanishing.

However, if we drop the classes based on purely random Bernoulli trials, we may exclude the classes that are important for classification. Obviously, the target class $t$ of a given instance should not be dropped, but we cannot manually set the retain probabilities $\rho_t = 1$ since the target classes differ for each instance, and more importantly, we do not know them at test time. We also want the retain probabilities $\rho_1, \ldots, \rho_K$ to encode meaningful correlations between classes, so that the highly correlated classes may be dropped or retained together to limit the hypothesis space to a meaningful subspace.

To resolve these issues, we adopt the idea of Adaptive Dropout [2], and model $\boldsymbol{\rho} \in [0, 1]^K$ as an output of a neural network which takes the last feature vector $\mathbf{h}$ as an input:

$$\boldsymbol{\rho}(\mathbf{x}; \theta) = \mathrm{sgm}(\mathbf{W}_\theta^\top \mathbf{h} + \mathbf{b}_\theta), \quad \theta = \{\mathbf{W}_\theta, \mathbf{b}_\theta\}. \tag{4}$$

By learning $\theta$, we expect these retain probabilities to be high for the target class of given inputs, and consider correlations between classes. Based on this retain probability network, DropMax is defined as follows.

$$z_k|\mathbf{x} \sim \mathrm{Ber}(z_k; \rho_k(\mathbf{x}; \theta)), \quad p(\mathbf{y}|\mathbf{x}, \mathbf{z}; \psi, \theta) = \frac{(z_t + \varepsilon) \exp(o_t(\mathbf{x}; \psi))}{\sum_k (z_k + \varepsilon) \exp(o_k(\mathbf{x}; \psi))} \tag{5}$$

The main difference of our model from [2] is that, unlike in the adaptive dropout where the neurons of intermediate layers are dropped, we drop *classes*. As we stated earlier, this is a critical difference, because by dropping classes we let the model to learn on different *(sub)-problems* at each iteration, while in the adaptive dropout we train different *models* at each iteration. Of course, our model can be extended to let it learn the dropout probabilities for the intermediate layers, but it is not our primary concern at this point. Note that DropMax can be easily applied to *any* type of neural networks, such as convolutional neural nets or recurrent neural nets, provided that they have the softmax output for the last layer. This generality is another benefit of our approach compared to the (adaptive) dropout that are reported to degrade the performance when used in the intermediate layers of convolutional or recurrent neural networks without careful configuration.

A limitation of [2] is the use of heuristics to learn the dropout probabilities that may possibly result in high variance in gradients during training. To overcome this weakness, we use concrete distribution [21], which is a continuous relaxation of discrete random variables that allows to back-propagate through the (relaxed) bernoulli random variables $z_k$ to compute the gradients of $\theta$ [9]:

$$z_k = \mathrm{sgm}\left\{\tau^{-1}\left(\log \rho_k(\mathbf{x}; \theta) - \log(1 - \rho_k(\mathbf{x}; \theta)) + \log u - \log(1 - u))\right)\right\} \tag{6}$$

with $u \sim \mathrm{Unif}(0,1)$. The temperature $\tau$ is usually set to 0.1, which determines the degree of probability mass concentration towards 0 and 1.

## 4 Approximate Inference for DropMax

In this section, we describe the learning framework for DropMax. For notational simplicity, we define $\mathbf{X}, \mathbf{Y}, \mathbf{Z}$ as the concatenations of $\mathbf{x}_i, \mathbf{y}_i$, and $\mathbf{z}_i$ over all training instances ($i = 1, \ldots, N$).

### 4.1 Intractable true posterior

We first check the form of the true posterior distribution $p(\mathbf{Z}|\mathbf{X}, \mathbf{Y}) = \prod_{i=1}^N p(\mathbf{z}_i|\mathbf{x}_i, \mathbf{y}_i)$. If it is tractable, then we can use exact inference algorithms such as EM to directly maximize the log-likelihood of our observation $\mathbf{Y}|\mathbf{X}$. For each instance, the posterior distribution can be written as

$$p(\mathbf{z}|\mathbf{x}, \mathbf{y}) = \frac{p(\mathbf{y}, \mathbf{z}|\mathbf{x})}{p(\mathbf{y}|\mathbf{x})} = \frac{p(\mathbf{y}|\mathbf{x}, \mathbf{z})p(\mathbf{z}|\mathbf{x})}{\sum_{\mathbf{z}'} p(\mathbf{y}|\mathbf{x}, \mathbf{z}')p(\mathbf{z}'|\mathbf{x})} \tag{7}$$

where we let $p(\mathbf{z}|\mathbf{x}) = \prod_{k=1}^K p(z_k|\mathbf{x})$ for simplicity. However, the graphical representation of (5) indicates the dependencies among $z_1, \ldots, z_K$ when $\mathbf{y}$ is observed. It means that unlike $p(\mathbf{z}|\mathbf{x})$, the true posterior $p(\mathbf{z}|\mathbf{x}, \mathbf{y})$ is not decomposable into the product of each element. Further, the denominator is the summation w.r.t. the exponentially many combinations of $\mathbf{z}$, which makes the form of the true posterior even more complicated.

Thus, we suggest to use stochastic gradient variational Bayes (SGVB), which is a general framework for approximating intractable posterior of latent variables in neural network [15, 24]. In standard variational inference, we maximize the evidence lower bound (ELBO):

$$\log p(\mathbf{Y}|\mathbf{X}; \psi, \theta) \geq \sum_{i=1}^N \left\{ \mathbb{E}_{q(\mathbf{z}_i|\mathbf{x}_i, \mathbf{y}_i; \phi)}\left[\log p(\mathbf{y}_i|\mathbf{z}_i, \mathbf{x}_i; \psi)\right] - \mathrm{KL}\left[q(\mathbf{z}_i|\mathbf{x}_i, \mathbf{y}_i; \phi)\big\|p(\mathbf{z}_i|\mathbf{x}_i; \theta)\right]\right\} \tag{8}$$

where $q(\mathbf{z}_i|\mathbf{x}_i, \mathbf{y}_i; \phi)$ is our approximate posterior with a set of variational parameters $\phi$.

### 4.2 Structural form of the approximate posterior

The probabilistic interpretation of each term in (8) is straightforward. However, it does not tell us how to encode them into the network components. Especially, in modeling $q(\mathbf{z}|\mathbf{x}, \mathbf{y}; \phi)$, how to utilize

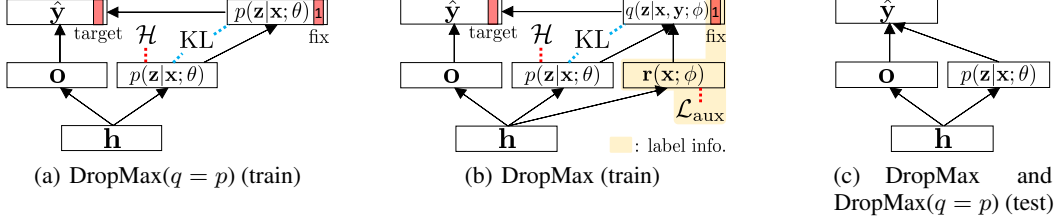

(a) DropMax($q = p$) (train)　　(b) DropMax (train)　　(c) DropMax and DropMax($q = p$) (test)

Figure 2: Illustration of model architectures. (a) DropMax ($q = p$) model at training time that lets $q(\mathbf{z}|\mathbf{x}, \mathbf{y}) = p(\mathbf{z}|\mathbf{x}, \mathbf{y}; \theta)$, except that it fixes the target mask as 1. (b) DropMax model that utilizes the label information at training time. (c) The test-time architecture for both models.

the label $\mathbf{y}$ is not a straightforward matter. [24] suggests a simple concatenation $[\mathbf{h}(\mathbf{x}); \mathbf{y}]$ as an input to $q(\mathbf{z}|\mathbf{x}, \mathbf{y}; \phi)$, while generating a pseudo-label from $p(\mathbf{z}|\mathbf{x}; \theta)$ to make the pipeline of training and testing network to be identical. However, it significantly increases the number of parameters of the whole network. On the other hand, [24] also proposes another solution where the approximate posterior simply ignores $\mathbf{y}$ and share the parameters with $p(\mathbf{z}|\mathbf{x}; \theta)$ (Figure 2(a)). This method is known to be stable due to the consistency between training and testing pipeline [24, 28, 13]. However, we empirically found that this approach produces suboptimal results for DropMax since it yields inaccurate approximated posterior.

Our novel approach to solve the problem starts from the observation that the relationship between $\mathbf{z}$ and $\mathbf{y}$ is relatively simple in DropMax (5), unlike the case where latent variables are assumed at lower layers. In this case, even though a closed form of true posterior is not available, we can capture a few important property of it and encode them into the approximate posterior.

The first step is to encode the structural form of the true posterior (7), which is decomposable into two factors: 1) the factor dependent only on $\mathbf{x}$, and 2) the factor dependent on both $\mathbf{x}$ and $\mathbf{y}$.

$$p(\mathbf{z}|\mathbf{x}, \mathbf{y}) = \underbrace{p(\mathbf{z}|\mathbf{x})}_{\mathcal{A}} \times \underbrace{p(\mathbf{y}|\mathbf{z}, \mathbf{x})/p(\mathbf{y}|\mathbf{x})}_{\mathcal{B}}. \tag{9}$$

The key idea is that the factor $\mathcal{B}$ can be interpreted as the rescaling factor from the unlabeled posterior $p(\mathbf{z}|\mathbf{x})$, which takes $\mathbf{x}$ and $\mathbf{y}$ as inputs. In doing so, we model the approximate posterior $q(\mathbf{z}|\mathbf{x}, \mathbf{y})$ with two pipelines. Each of them corresponds to: $\mathcal{A}$ without label, which we already have defined as $p(\mathbf{z}|\mathbf{x}; \theta) = \mathrm{Ber}(\mathbf{z}; \mathrm{sgm}(\mathbf{W}_\theta^\top \mathbf{h} + \mathbf{b}_\theta))$ in (4), and $\mathcal{B}$ with label, which we discuss below.

$\mathcal{B}$ is able to scale up or down $\mathcal{A}$, but at the same time should bound the resultant posterior $p(\mathbf{z}|\mathbf{x}, \mathbf{y})$ in the range of $[0, 1]^K$. To model $\mathcal{B}$ with network components, we simply add to the logit of $\mathcal{A}$, a vector $\mathbf{r} \in \mathbb{R}^K$ taking $\mathbf{x}$ as an input (we will defer the description on how to use $\mathbf{y}$ to the next subsection). Then we squash it again in the range of $[0, 1]$ (Note that addition in the logit level is multiplicative):

$$\mathbf{g}(\mathbf{x}; \phi) = \mathrm{sgm}(\overline{\mathbf{W}}_\theta^\top \mathbf{h} + \overline{\mathbf{b}}_\theta + \mathbf{r}(\mathbf{x}; \phi)), \quad \mathbf{r}(\mathbf{x}; \phi) = \mathbf{W}_\phi^\top \mathbf{h} + \mathbf{b}_\phi. \tag{10}$$

where $\mathbf{g}(\mathbf{x}; \phi)$ is the main ingredient of our approximate posterior in (12), and $\phi = \{\mathbf{W}_\phi, \mathbf{b}_\phi\}$ is variational parameter. $\overline{\mathbf{W}}_\theta$ and $\overline{\mathbf{b}}_\theta$ denote that stop-gradients are applied to them, to make sure $\mathbf{g}(\mathbf{x}; \phi)$ is only parameterized by the variational parameter $\phi$, which is important for properly definiting the variational distribution. Next we discuss how to encode $\mathbf{y}$ into $\mathbf{r}(\mathbf{x}; \phi)$ and $\mathbf{g}(\mathbf{x}; \phi)$, to finalize the approximate posterior $q(\mathbf{z}|\mathbf{x}, \mathbf{y}; \phi)$.

### 4.3　Encoding the label information

Our modeling choice for encoding $\mathbf{y}$ is based on the following observations.

**Observation 1.** *If we are given* $(\mathbf{x}, \mathbf{y})$ *and consider* $z_1, \ldots, z_K$ *one by one, then* $z_t$ *is positively correlated with the DropMax likelihood* $p(\mathbf{y}|\mathbf{x}, \mathbf{z})$ *in* (5)*, while* $z_{k \neq t}$ *is negatively correlated with it.*

**Observation 2.** *The true posterior of the target retain probability* $p(z_t = 1|\mathbf{x}, \mathbf{y})$ *is* 1*, if we exclude the case* $z_1 = z_2 = \cdots = z_K = 0$*, i.e. the retain probability for every class is* 0*.*

One can easily verify the observation 1: the likelihood will increase if we attend the target more, and vice versa. We encode this observation as follows. Noting that the likelihood $p(\mathbf{y}|\mathbf{x}, \mathbf{z})$ is in general maximized over the training instances, the factor $\mathcal{B}$ in (9) involves $p(\mathbf{y}|\mathbf{x}, \mathbf{z})$ and should behave

consistently (as in observation 1). Toward this, each $r_t(\mathbf{x}; \phi)$ and $r_{k \neq t}(\mathbf{x}; \phi)$ should be maximized and minimized respectively. We achieve this by minimizing the cross-entropy for $\mathrm{sgm}(\mathbf{r}(\mathbf{x}; \phi))$ across the training instances:

$$\mathcal{L}_{\mathrm{aux}}(\phi) = -\sum_{i=1}^{N} \sum_{k=1}^{K} \left\{ y_{i,k} \log \mathrm{sgm}(r_k(\mathbf{x}_i; \phi)) + (1 - y_{i,k}) \log(1 - \mathrm{sgm}(r_k(\mathbf{x}_i; \phi))) \right\} \quad (11)$$

The observation 2 says that $\mathbf{z}_{\backslash t} \neq \mathbf{0} \to z_t = 1$ given $\mathbf{y}$. Thus, simply ignoring the case $\mathbf{z}_{\backslash t} = \mathbf{0}$ and fixing $q(z_t|\mathbf{x}, \mathbf{y}; \phi) = \mathrm{Ber}(z_t; 1)$ is a close approximation of $p(z_t|\mathbf{x}, \mathbf{y})$, especially under mean-field assumption (see the Appendix A for justification). Hence, our final approximate posterior is given as:

$$q(\mathbf{z}|\mathbf{x}, \mathbf{y}; \phi) = \mathrm{Ber}(z_t; 1) \prod_{k \neq t} \mathrm{Ber}\left(z_k; g_k(\mathbf{x}; \phi)\right). \quad (12)$$

See Figure 2(b) and 2(c) for the illustration of the model architecture.

## 4.4 Regularized variational inference

One last critical issue in optimizing the ELBO (8) is that $p(\mathbf{z}|\mathbf{x}; \theta)$ collapses into $q(\mathbf{z}|\mathbf{x}, \mathbf{y}; \phi)$ too easily, as $p(\mathbf{z}|\mathbf{x}; \theta)$ is parameteric with input $\mathbf{x}$. Preventing it is crucial for $\mathbf{z}$ to generalize well on a test instance $\mathbf{x}^*$, because $\mathbf{z}$ is sampled from $p(\mathbf{z}|\mathbf{x}^*; \theta)$ at test time (Figure 2(c)). We empirically found that imposing some prior (e.g. zero-mean gaussian or laplace prior) to $\theta = \{\mathbf{W}_\theta, \mathbf{b}_\theta\}$ was not effective in preventing this behavior. (The situation is different from VAE [15] where the prior of latent code $p(\mathbf{z})$ is commonly set to gaussian with no trainable parameters (i.e. $\mathcal{N}(\mathbf{0}, \lambda \mathbf{I})$).)

We propose to remove weight decay for $\theta$ and apply an entropy regularizer directly to $p(\mathbf{z}|\mathbf{x}; \theta)$. We empirically found that the method works well without any scaling hyperparameters.

$$\mathcal{H}(p(\mathbf{z}|\mathbf{x}; \theta)) = \sum_k \rho_k \log \rho_k + (1 - \rho_k) \log(1 - \rho_k) \quad (13)$$

We are now equipped with all the essential components. The KL divergence and the final minimization objective are given as:

$$\mathrm{KL}[q(\mathbf{z}|\mathbf{x}, \mathbf{y}; \phi) || p(\mathbf{z}|\mathbf{x}; \theta)] = \sum_k \left\{ \mathbb{I}_{\{k=t\}} \log \frac{1}{\rho_k} + \mathbb{I}_{\{k \neq t\}} \left( g_k \log \frac{g_k}{\rho_k} + (1 - g_k) \log \frac{1 - g_k}{1 - \rho_k} \right) \right\}$$

$$\mathcal{L}(\psi, \theta, \phi) = \sum_{i=1}^{N} \left[ -\frac{1}{S} \sum_{s=1}^{S} \log p(y_i|\mathbf{x}_i, \mathbf{z}_i^{(s)}; \psi) + \mathrm{KL}[q(\mathbf{z}_i|\mathbf{x}_i, \mathbf{y}_i; \phi) || p(\mathbf{z}_i|\mathbf{x}_i; \theta)] - \mathcal{H} \right] + \mathcal{L}_{\mathrm{aux}}$$

where $\mathbf{z}_i^{(s)} \sim q(\mathbf{z}_i|\mathbf{x}_i, \mathbf{y}_i; \phi)$ and $S = 1$ as usual. Figure 2(b) and (c) illustrate the model architectures for training and testing respectively.

When testing, we can perform Monte-Carlo sampling:

$$p(\mathbf{y}^*|\mathbf{x}^*) = \mathbb{E}_{\mathbf{z}}[p(\mathbf{y}^*|\mathbf{x}^*, \mathbf{z})] \approx \frac{1}{S} \sum_{s=1}^{S} p(\mathbf{y}^*|\mathbf{x}^*, \mathbf{z}^{(s)}), \quad \mathbf{z}^{(s)} \sim p(\mathbf{z}|\mathbf{x}^*; \theta). \quad (14)$$

Alternatively, we can approximate the expectation as

$$p(\mathbf{y}^*|\mathbf{x}^*) = \mathbb{E}_{\mathbf{z}}[p(\mathbf{y}^*|\mathbf{x}^*, \mathbf{z})] \approx p(\mathbf{y}^*|\mathbf{x}^*, \mathbb{E}[\mathbf{z}|\mathbf{x}^*]) = p(\mathbf{y}^*|\mathbf{x}^*, \rho(\mathbf{x}^*; \theta)), \quad (15)$$

which is a common practice for many practitioners. We report test error based on (15).

# 5 Experiments

**Baselines and our models** We first introduce relevant baselines and our models.

**1) Base Softmax.** The baseline CNN network with softmax, that only uses the hidden unit dropout at fully connected layers, or no dropout regularization at all.

**2) Sparsemax.** Base network with Sparsemax loss proposed by [22], which produces sparse class probabilities.

Table 1: **Classification performance in test error (%).** The reported numbers are mean and standard errors with $95\%$ confidence interval over 5 runs.

| Models | M-$1K$ | M-$5K$ | M-$55K$ | C10 | C100 | AWA | CUB |
|---|---|---|---|---|---|---|---|
| Base Softmax | $7.09_{\pm 0.46}$ | $2.13_{\pm 0.21}$ | $0.65_{\pm 0.04}$ | $7.90_{\pm 0.21}$ | $30.60_{\pm 0.12}$ | $30.29_{\pm 0.80}$ | $48.84_{\pm 0.85}$ |
| Sparsemax [22] | $6.57_{\pm 0.17}$ | $2.05_{\pm 0.18}$ | $0.75_{\pm 0.06}$ | $7.90_{\pm 0.28}$ | $31.41_{\pm 0.16}$ | $36.06_{\pm 0.64}$ | $64.41_{\pm 1.12}$ |
| Sampled Softmax [12] | $7.36_{\pm 0.22}$ | $2.31_{\pm 0.14}$ | $0.66_{\pm 0.04}$ | $7.98_{\pm 0.24}$ | $30.87_{\pm 0.19}$ | $29.81_{\pm 0.45}$ | $49.90_{\pm 0.56}$ |
| Random DropMax | $7.19_{\pm 0.57}$ | $2.23_{\pm 0.19}$ | $0.68_{\pm 0.07}$ | $8.21_{\pm 0.08}$ | $30.78_{\pm 0.28}$ | $31.11_{\pm 0.54}$ | $48.87_{\pm 0.79}$ |
| Deterministic Attention | $6.91_{\pm 0.46}$ | $2.03_{\pm 0.11}$ | $0.69_{\pm 0.05}$ | $7.87_{\pm 0.24}$ | $30.60_{\pm 0.21}$ | $30.98_{\pm 0.66}$ | $49.97_{\pm 0.32}$ |
| Deterministic DropMax | $6.30_{\pm 0.64}$ | $1.89_{\pm 0.04}$ | $0.64_{\pm 0.05}$ | $7.84_{\pm 0.14}$ | $30.55_{\pm 0.51}$ | $\mathbf{26.22_{\pm 0.76}}$ | $47.35_{\pm 0.42}$ |
| DropMax ($q = p$) | $7.52_{\pm 0.26}$ | $2.05_{\pm 0.07}$ | $0.63_{\pm 0.02}$ | $7.80_{\pm 0.22}$ | $29.98_{\pm 0.35}$ | $29.27_{\pm 1.19}$ | $42.08_{\pm 0.94}$ |
| DropMax | $\mathbf{5.32_{\pm 0.09}}$ | $\mathbf{1.64_{\pm 0.08}}$ | $\mathbf{0.59_{\pm 0.04}}$ | $\mathbf{7.67_{\pm 0.11}}$ | $\mathbf{29.87_{\pm 0.36}}$ | $26.91_{\pm 0.54}$ | $\mathbf{41.07_{\pm 0.57}}$ |

**3) Sampled Softmax.** Base network with sampled softmax [12]. Sampling function $Q(\mathbf{y}|\mathbf{x})$ is uniformly distributed during training. We tune the number of sampled classes among $\{20\%, 40\%, 60\%\}$ of total classes, while the target class is always selected. Test is done with (2).

**4) Random DropMax.** A baseline that randomly drops out non-target classes with a predefined retain probability $\rho \in \{0.2, 0.4, 0.6\}$ at training time. For learning stability, the target class is not dropped out during training. Test is done with the softmax function (2), without sampling the dropout masks.

**5) Deterministic Attention.** Softmax with deterministic sigmoid attentions multiplied at the exponentiations. The attention probabilities are generated from the last feature vector $\mathbf{h}$ with additional weights and biases, similarly to (4).

**6) Deterministic DropMax.** This model is the same with Deterministic Attention, except that it is trained in a supervised manner with true class labels. With such consideration of labels when training the attention generating network, the model can be viewed as a deterministic version of DropMax.

**7) DropMax ($q = p$).** A variant of DropMax where we let $q(\mathbf{z}|\mathbf{x}, \mathbf{y}) = p(\mathbf{z}|\mathbf{x}; \theta)$ except that we fix $q(z_t|\mathbf{x}, \mathbf{y}) = \mathrm{Ber}(z_t; 1)$ as in (12) for learning stability. The corresponding $\mathrm{KL}[q\|p]$ can be easily computed. The auxiliary loss term $\mathcal{L}_{\mathrm{aux}}$ is removed and the entropy term $\mathcal{H}$ is scaled with a hyperparameter $\gamma \in \{1, 0.1, 0.01\}$ (See Figure 2(a)).

**8) DropMax.** Our adaptive stochastic softmax, where each class is dropped out with input dependent probabilities trained from the data. No hyperparameters are needed for scaling each term.

We implemented DropMax using Tensorflow [1] framework. The source codes are available at `https://github.com/haebeom-lee/dropmax`.

**Datasets and base networks**    We validate our method on multiple public datasets for classification, with different network architecture for each dataset.

**1) MNIST.** This dataset [19] consists of $60,000$ images that describe hand-written digits from 0 to 9. We experiment with varying number of training instances: $1K$, $5K$, and $55K$. The validation and test set has $5K$ and $10K$ instances, respectively. As for the base network, we use the CNN provided in the Tensorflow Tutorial, which has a similar structure to LeNet.

**2) CIFAR-10.** This dataset [16] consists of 10 generic object classes, which for each class has 5000 images for training and 1000 images for test. We use ResNet-34 [10] as the base network.

**3) CIFAR-100.** This dataset consists of 100 object classes. It has 500 images for training and 100 images are for test for each class. We use ResNet-34 as the base network.

**4) AWA.** This is a dataset for classifying different animal species [18], that contains $30,475$ images from 50 animal classes such as *cow*, *fox*, and *humpback whale*. For each class, we used 50 images for test, while rest of the images are used as training set. We use ResNet-18 as the base network.

**5) CUB-200-2011.** This dataset [26] consists of 200 bird classes such as *Black footed albatross*, *Rusty blackbird*, and *Eastern towhee*. It has 5994 training images and 5794 test images, which is quite small compared to the number of classes. We only use the class label for the classification. We use ResNet-18 as the base network.

As AWA and CUB datasets are subsets of ImageNet-1K dataset, for those datasets we do not use a pretained model but train from scratch. The experimental setup is available in the Appendix C.

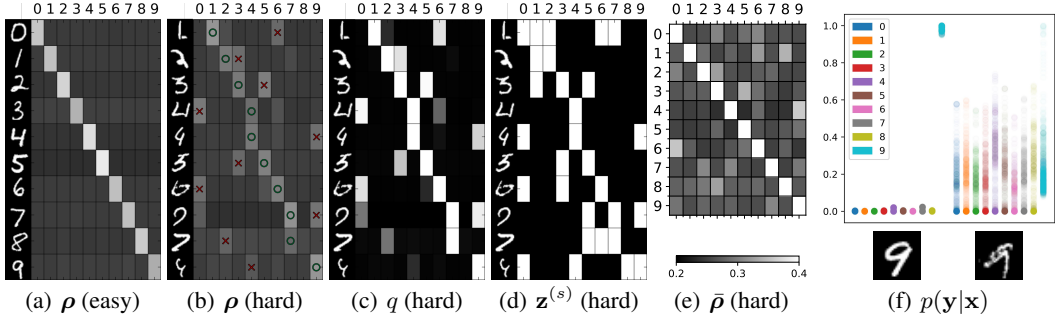

(a) $\boldsymbol{\rho}$ (easy)    (b) $\boldsymbol{\rho}$ (hard)    (c) $q$ (hard)    (d) $\mathbf{z}^{(s)}$ (hard)    (e) $\bar{\boldsymbol{\rho}}$ (hard)    (f) $p(\mathbf{y}|\mathbf{x})$

Figure 4: Visualization of class dropout probabilities for example test instances from MNIST-1K dataset. (a) and (b) shows estimated class retain probability for easy and difficult test instances respectively. The green o's denote the ground truths, while the red x's denote the base model predictoins. (c) shows approximate posterior $q(\mathbf{z}|\mathbf{x}, \mathbf{y}; \phi)$. (d) shows generated retain masks from (b). (e) shows the average retain probability per class for hard instances. (f) shows sampled predictive distributions of easy and difficult instance respectively.

## 5.1 Quantitative Evaluation

**Multi-class classification.** We report the classification performances of our models and the baselines in Table 1. The results show that the variants of softmax function such as Sparsemax and Sampled Softmax perform similarly to the original softmax function (or worse). Random DropMax also performs worse due to the inconsistency between train and test time handling of the dropout probabilities for target class. Deterministic Attention also performs similarly to all the previous baselines. Interestingly, Deterministic DropMax with supervised learning of attention mechanism improves the performance over the base soft classifier, which suggests that such combination of a multi-class and multi-label classifier could be somewhat beneficial. However, the improvements are marginal except on the AWA dataset, because the gating function also lacks proper regularization add thus yields very sharp attention probabilities. DropMax $(q = p)$ has an entropy regularizer to address this issue, but the model obtains suboptimal performance due to the inaccurate posterior estimation.

On the other hand, the gating function of DropMax is optimally regularized to make a *crude* selection of candidate classes via the proposed variational inference framework, and shows consistent and significant improvements over the baselines across all datasets. DropMax also obtains noticeably higher accuracy gains on AWA and CUB dataset that are fine-grained, with $3.38\%p$ and $7.77\%p$ improvements, as these fine-grained datasets contain many ambiguous instances that can be effectively handled by DropMax with its focus on the most confusing classes. On MNIST dataset, we also observe that the DropMax is more effective when the number of training instances is small. We attribute this to the effect of stochastic regularization that effectively prevents overfitting. This is also a general advantage of Bayesian learning as well.

**Convergence rate.** We examine the convergence rate of our model against the base network with regular softmax function. Figure 3 shows the plots of cross entropy loss computed at each training step on MNIST-55K and CIFAR-100. To reduce the variance of $\mathbf{z}$, we plot with $\boldsymbol{\rho}$ instead (training is done with $\mathbf{z}$). DropMax shows slightly lower convergence rate, but the test loss is significantly improved, effectively preventing overfitting. Moreover, the learning curve of DropMax is more stable than that of regular softmax (see Appendix B for more discussion).

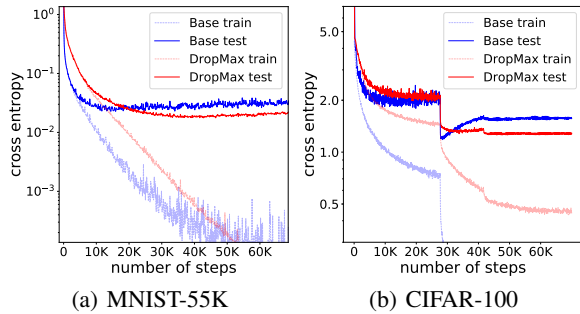

(a) MNIST-55K      (b) CIFAR-100

Figure 3: Convergence plots

## 5.2 Qualitative Analysis

We further perform qualitative analysis of our model to see how exactly it works and where the accuracy improvements come from.

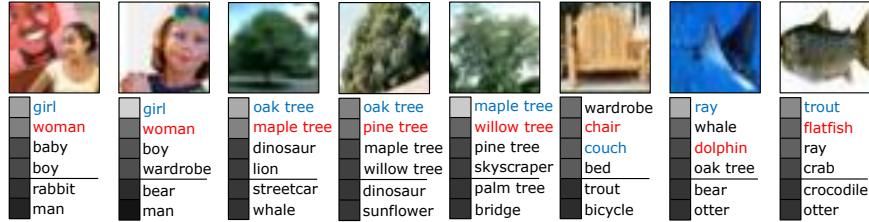

Figure 5: Examples from CIFAR-100 dataset with top-4 and bottom-2 retain probabilities. Blue and red color denotes the ground truths and base model predictions respectively.

Figure 4(a) shows the retain probabilities estimated for easy examples, in which case the model set the retain probability to be high for the true class, and evenly low for non-target classes. Thus, when the examples are easy, the dropout probability estimator works like a second classifier. However, for difficult examples in Figure 4(b) that is missclassified by the base softmax function, we observe that the retain probability is set high for the target class and few other candidates, as this helps the model focus on the classification between them. For example, in Figure 4(b), the instance from class 3 sets high retain probability for class 5, since its handwritten character looks somewhat similar to number 5. However, the retain probability could be set differently even across the instances from the same class, which makes sense since even within the same class, different instances may get confused with different classes. For example, for the first instance of 4, the class with high retain probability is 0, which looks like 0 in its contour. However, for the second instance of 4, the network sets class 9 with high retain probability as this instance looks like 9.

Similar behaviors can be observed on CIFAR-100 dataset (Figure 5) as well. As an example, for instances that belong to class *girl*, DropMax sets the retain probability high on class *woman* and *boy*, which shows that it attends to most confusing classes to focus more on such difficult problems.

We further examine the class-average dropout probabilities for each class in MNIST dataset in Figure 4(e). We observe the patterns by which classes are easily confused with the others. For example, class 3 is often confused with 5, and class 4 with 9. It suggests that retain probability implicitly learns correlations between classes, since it is modeled as an input dependent distribution. Also, since DropMax is a Bayesian inference framework, we can easily obtain predictive uncertainty from MC sampling in Figure 4(f), even when probabilistic modeling on intermediate layers is difficult.

## 6   Conclusion and Future Work

We proposed a stochastic version of a softmax function, DropMax, that randomly drops non-target classes at each iteration of the training step. DropMax enables to build an ensemble over exponentially many classifiers that provide different decision boundaries. We further proposed to learn the class dropout probabilities based on the input, such that it can consider the discrimination of each instance against more confusing classes. We cast this as a Bayesian learning problem and present how to optimize the parameters through variational inference, while proposing a novel regularizer to more exactly estimate the true posterior. We validate our model on multiple public datasets for classification, on which our model consistently obtains significant performance improvements over the base softmax classifier and its variants, achieving especially high accuracy on datasets for fine-grained classification. For future work, we plan to further investigate the source of generalization improvements with DropMax, besides increased stability of gradients (Appendix B).

### Acknowledgement

This research was supported by the Engineering Research Center Program through the National Research Foundation of Korea (NRF) funded by the Korean Government MSIT (NRF-2018R1A5A1059921), Samsung Research Funding Center of Samsung Electronics (SRFC-IT150203), Machine Learning and Statistical Inference Framework for Explainable Artificial Intelligence (No.2017-0-01779), and Basic Science Research Program through the National Research Foundation of Korea (NRF) funded by the Ministry of Education (2015R1D1A1A01061019). Juho Lee's research leading to these results has received funding from the European Research Council under the European Union's Seventh Framework Programme (FP7/2007-2013) ERC grant agreement no. 617071.

## Footnotes

[1]to number of classes

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
