[Supplementary Material · supplementary.pdf]

# Appendix for DropMax: Adaptive Variational Softmax

**Hae Beom Lee**[1,2]**, Juho Lee**[3,2]**, Saehoon Kim**[2]**, Eunho Yang**[1,2]**, Sung Ju Hwang**[1,2]
KAIST[1], AItrics[2], South Korea,
University of Oxford[3], United Kingdom,
{haebeom.lee, eunhoy, sjhwang82}@kaist.ac.kr
juho.lee@stats.ox.ac.uk, shkim@aitrics.com

## A  Justification of the Observation 2.

Here we provide the justification and intuition of the observation 2 in Section 4.3 of the main paper.

**Observation 2.**   *The true posterior of the target retain probability $p(z_t = 1|\mathbf{x}, \mathbf{y})$ is 1, if we exclude the case $z_1 = z_2 = \cdots = z_K = 0$, i.e. the retain probability for every class is 0.*

To verify it, we first need to understand what it means by saying $z_t = 0$ even after the observation of the target $\mathbf{y}$. Firstly, suppose that the target mask $z_t = 0$ and there exists at least one nontarget mask $z_{j \neq t} = 1$. Then, the corresponding likelihood and the true posterior becomes

$$p(\mathbf{y}|\mathbf{x}, z_t = 0, \mathbf{z}_{\backslash t}) = \frac{(0 + \varepsilon)\exp(o_t)}{(1 + \varepsilon)\exp(o_j) + \sum_{k \neq j}(z_k + \varepsilon)\exp(o_k)} \approx 0 \tag{1}$$

$$p(z_t = 0, \mathbf{z}_{\backslash t}|\mathbf{x}, \mathbf{y}) = p(\mathbf{y}|\mathbf{x}, z_t = 0, \mathbf{z}_{\backslash t})\frac{p(z_t = 0, \mathbf{z}_{\backslash t}|\mathbf{x})}{p(\mathbf{y}|\mathbf{x})} \approx 0 \tag{2}$$

where $\varepsilon > 0$ is a sufficiently small constant (e.g. $10^{-20}$). In other words, after knowing which class is the target, it is impossible to reason that the target class has been dropped out while some nontarget classes have not.

Secondly, suppose $z_t = 0$ and $\mathbf{z}_{\backslash t} = \mathbf{0}$. Then the likelihood and the true posterior becomes

$$p(y_t = 1|\mathbf{x}, \mathbf{z} = \mathbf{0}) = \frac{(0 + \varepsilon)\exp(o_t)}{\sum_k(0 + \varepsilon)\exp(o_k)} = \frac{\exp(o_t)}{\sum_k \exp(o_k)} > 0 \tag{3}$$

$$p(\mathbf{z} = \mathbf{0}|\mathbf{x}, \mathbf{y}) = p(\mathbf{y}|\mathbf{x}, \mathbf{z} = \mathbf{0})\frac{p(\mathbf{z} = \mathbf{0}|\mathbf{x})}{p(\mathbf{y}|\mathbf{x})} \geq 0 \tag{4}$$

In other words, after observing the label, it is one of the possible scenarios that all the target and nontarget classs have been dropped out at the same time. Combining (2) and (4), we can conclude that $z_t = 0$ only if $\mathbf{z}_{\backslash t} = \mathbf{0}$, given $\mathbf{y}$. Ohterwise, $z_t = 1$.

Then, how can we express this relationship with the approximate posterior $q(\mathbf{z}|\mathbf{x}, \mathbf{y}) = \prod_k q(z_k|\mathbf{x}, \mathbf{y})$? It is impossible because we do not consider the correlations between $z_1, \ldots, z_K$ under the mean-field approximation. In such a case, if we allow $q(z_t|\mathbf{x}, \mathbf{y}) < 1$ somehow while having no means to force $z_t = 0 \to \mathbf{z}_{\backslash t} = \mathbf{0}$, then whenever $z_t$ is realized to be 0, we always see the devation from the true posterior by the amount $q(\mathbf{z}_{\backslash t}|\mathbf{x}, \mathbf{y})$ deviates from $\prod_{k \neq t} \text{Ber}(z_k; 0)$. It also causes severe learning instability since reverting $z_t$ back to 1 requires huge gradients. Considering that the case $\mathbf{z}_{\backslash t} = \mathbf{0}$, one of the $2^{K-1}$ combinations, is insignificant, we ignore this case and let $q(z_t|\mathbf{x}, \mathbf{y}) = \text{Ber}(z_t; 1)$. Except that case, the solution exactly matches the true marginal posterior $p(z_t|\mathbf{x}, \mathbf{y})$.

# B Stability of Gradients

The effect of DropMax regularization can be also explained in the context of the stability of stochastic gradient descent (SGD) [1, 2], where a stable algorithm is preferred to achieve small generalization error. Suppose that the current model correctly classifies an example with small confidence. DropMax regularization incurs a penalty to restrict the model from classifying an example too much perfectly (i.e. $o_t \gg \max_{k \in [K] \setminus \{t\}} o_k$). This automatically suggests that the magnitude of gradients of DropMax at this example is smaller than that of softmax, which helps to prevent from over-fitting and generalize better, as discussed in [2].

Denoting $\psi = \{\mathbf{W}, \mathbf{b}\}$, we consider the expected cross entropy as our loss function:

$$\sum_{i=1}^{N} l(\mathbf{x}_i, \mathbf{y}_i; \psi) = \sum_{i=1}^{N} \mathbb{E}_{\mathbf{z}_i} \left[ - \log p\left(y_{i,t} = 1 | \mathbf{h}_i, \mathbf{z}_i; \psi\right) \right], \tag{5}$$

where $\mathbf{h}_i = \mathrm{NN}(\mathbf{x}_i; \omega)$ is the last feature vector of an arbitrary neural network, $\mathbf{z}_i$ and $p(y_{i,t} = 1 | \mathbf{h}_i, \mathbf{z}_i; \psi)$ are defined in Eq. (5) in the main paper. We consider an example that is correctly classified with small confidence;

**Condition 1.** *Suppose that we are given a labeled example* $\mathbf{x}_i$ *and* $\mathbf{y}_i$. *We assume that the retain probabilities denoted by* $\{\rho_k\}_{k=1}^{K}$ *follow the case: For a target class* $t$, $\rho_t$ *is greater than* $\max_{k \in [K] \setminus \{t\}} \rho_k$. *For a non-target class* $k$, $\rho_k$ *is equal to the one of any non-target classes.*

We further assume that Bernoulli parameter for $\mathbf{z}_i$ is fixed, but different for each example. For simplicity, we denote $o_k(\mathbf{x}_i; \psi)$ as $o_k$ when the context is clear.

We then decompose the expected loss into the standard cross entropy with softmax and the regularization term introduced by DropMax;

$$\sum_{i=1}^{N} \left( \widehat{l}(\mathbf{x}_i, \mathbf{y}_i; \psi) + \mathcal{M}(\mathbf{x}_i, \mathbf{y}_i, \mathbf{z}_i) \right), \tag{6}$$

where $\widehat{l}(\mathbf{x}_i, \mathbf{y}_i; \psi) = - \log \frac{\exp(o_t)}{\sum_{k=1}^{K} \exp(o_k)}$ that is the standard cross-entropy loss with softmax and $\mathcal{M}(\mathbf{x}_i, \mathbf{y}_i, \mathbf{z}_i) = \mathbb{E}_{\mathbf{z}_i} \left[ \log \frac{\sum_{k=1}^{K} (z_k + \epsilon) \exp(o_k)}{(z_t + \epsilon) \sum_{k=1}^{K} \exp(o_k)} \right]$. We derive the upper bound on the regularization term by Jensen's inequality and keep terms only related to $\psi$;

$$\sum_{i=1}^{N} \left[ \log \sum_{k=1}^{K} (\rho_k + \epsilon) \exp(o_k) - \log \sum_{k=1}^{K} (\exp(o_k)) \right] \tag{7}$$

We now compute the magnitude of gradient of DropMax to show if it is smaller than the one of softmax, which helps to stabilize the learning procedure. For ease of analysis, we consider the gradient for a target class[1]:

$$\frac{\partial \mathcal{M}(\mathbf{x}_i, \mathbf{y}_i, \mathbf{z}_i)}{\partial \mathbf{w}_t} \leq \left( \frac{(\rho_t + \epsilon) \exp(o_t)}{\sum_k (\rho_k + \epsilon) \exp(o_k)} - \frac{\exp(o_t)}{\sum_k \exp(o_k)} \right) \frac{\partial o_t}{\partial \mathbf{w}_t} \tag{8}$$

$$\frac{\partial \widehat{l}(\mathbf{x}_i, \mathbf{y}_i; \psi)}{\partial \mathbf{w}_t} = \left( \frac{\exp(o_t)}{\sum_k \exp(o_k)} - 1 \right) \frac{\partial o_t}{\partial \mathbf{w}_t}. \tag{9}$$

According to Condition 1, it is easy to see that

$$0 < \left( \frac{(\rho_t + \epsilon) \exp(o_t)}{\sum_k (\rho_k + \epsilon) \exp(o_k)} - \frac{\exp(o_t)}{\sum_k \exp(o_k)} \right), \tag{10}$$

which suggests that the gradient direction of regularizer is opposite to that of $\widehat{l}(\mathbf{x}_i, \mathbf{y}_i; \psi)$. For an example that can be correctly classified with small margin, DropMax regularization incurs a penalty to restrict the model from classifying an example too much perfectly (i.e. $o_t \gg \max_{k \in [K] \setminus \{t\}} o_k$). This means that DropMax is relatively more stable than softmax in the notion of magnitude of gradient, which helps to prevent from over-fitting and generalize better.

Figure 1: Contour plots of softmax and DropMax with different retain probabilities. For DropMax, we sampled the Bernoulli variables for each data point with fixed probabilities.

Figure 2: (a) Monte-Carlo sampling of the target probabilities ($S = 1000$) w.r.t. the different amount of noise on an instance from class 1. (b) Same as (a), except we do not sample the target mask to reduce the unnecessary variances (simply replace $z_t \sim \text{Ber}(\rho_t)$ with $\rho_t$). (c) MC sampling with real examples having different level of difficulties.

The convergence plot of MNIST-55K dataset (Figure 3(a) in the main paper) supports agrees with our argument that DropMax generalizes better by improving the stability of learning. Once the retain probabilies are trained to some degree and can roughly classify target and nontarget classes with minimum risk, then the burden to the softmax classifier is lessened, resulting in more stable gradients for the main softmax classifier.

## C  Experimental Setup

Here we explain the experimental setup for the each dataset.

**1) MNIST.** The batchsize is set to $50$ and the training epoch is set to $2000$, $500$, and $100$ for $1K$,$5K$, and $55K$ dataset, respectively. We use Adam optimizer [3], with learning rate starting from $10^{-4}$. The $\ell_2$ weight decay parameter is searched in the range of $\{0, 10^{-5}, 10^{-4}, 10^{-3}\}$. All the hyperparameters are tuned with a holdout set.

**2) CIFAR-10.** We set batchsize to $128$ and the number of training epoch to $200$. We use stochastic gradient descent (SGD) optimizer with $0.9$ momentum. Learning rate starts from $0.1$ and multiplied by $0.1$ at $80, 120, 160$ epochs. The $\ell_2$ weight decay parameter is fixed at $10^{-4}$.

**3) CIFAR-100.** We used the same setup as CIFAR-10.

**4) AWA.** Batchsize is set to $125$ and the number of training epochs is set to $300$. We use SGD optimizer with $0.9$ momentum. Learning rate starts from $10^{-2}$, and is multiplied by $0.1$ at $150$ and $250$ epochs. Weight decay is set to $10^{-4}$.

**5) CUB-200-2011.** Batchsize is set to $125$ and the number of training epochs is set to $400$. SGD optimizer with $0.9$ momentum is used. Learning rate starts from $10^{-2}$ and is multiplied by $0.1$ at $200$ and $300$ epochs. We set the weight decay to $10^{-3}$ which is bigger than the other datasets, considering that the size of the dataset is small compared to the network capacity.

## Footnotes

[1]We can make the similar arguments for non-target classes.