[Reviews · NeurIPS 2018]

Reviewer 1



On the very high level, I am not convinced with the motivation. It seems the motivation is solely to focus “more” on the confusing classes for each sample. - Isn’t this already happening when using softmax? That is, the backpropagation gradient of softmax-cross entropy is proportional to the difference of true and predicted probabilities. In the case of this work, the extent of focus is obtained by “another predictor” which acts the same as a class predictor as far as I understand. This first brings me to the second question, - why do we need another affine transform to obtain \rho and not use the class probabilities in the first place? Especially in light of added cross entropy loss in (11). The experimental results and the comparison to baselines show improved performance on multiple datasets. However, since most state-of-the-art results on benchmark datasets use softmax, it would have been more convincing if better results could be achieved using those architectures and the set of hyper parameters. For instance on ImageNet and one of the top recent architectures. Also, other quantitative and qualitative results do not show any insight beyond that of a standard softmax and the confusion matrix and predictive uncertainty obtained from it. The rebuttal provides more baselines which better shows the significance of the approach.

Reviewer 2



Paper is concise overall. It has thrown new light on variational dropout with clear literature survey and step by step delineation of its ideas. This paper uses a novel combination of variational dropouts, KL divergence to do variational inferencing in a two step process. First a crude selection of candidate classes is achieved by automatically dropping out the wrong classes using a neural network. The second step, involving using a standard softmax, is simpler because the network has to attend to only a few classes while classification. The dropout probabilities are learned in a novel way, by using a graph that combines target, input, 2 different neural networks that learn dropout-layers. In the two networks, one uses the target to prioritize non-dropping of target while the other network tries to mimic the first network via minimizing KL-Divergence. This is a necessary step as testing phase doesn't involve a target. The experiments are presented, although the setup could've been more detailed (choice of optimizer, number of iterations etc...). The explanation of the results using plots is clear, and so are the explanations in Qualitative Analysis section. Interesting thing to notice is, the approach getting better results (compared to other standard techniques) on smaller datasets.

Reviewer 3



This paper proposes doing dropout in the output softmax layer during supervised training of neural net classifiers. The dropout probabilities are adapted per example. The probabilities are computed as a function of the penultimate layer of the classifier. So that layer is used to compute both : the logits, and the gating for those logits. This model combines ideas from adaptive dropout (Ba and Frey NIPS'13) and variational dropout (Kingma et al). The key problem being solved is how to do inference to get the optimal dropout probabilities. There are several approximations made in the model in order to do this inference. If I understood correctly, the model essentially comes down to the following: Two gating functions are being imposed to gate the logits. One gating function is computed by taking the penultimate layer and passing it through learned weights (W_theta). The other gating function is also computed by taking the penultimate layer and passing it through learned weights (W_phi), but the second gating also tries to take the output y into account by regressing to it using an auxiliary loss. Together the two gating functions produce a number in [0, 1] which is used as the retain probability for doing dropout in the output softmax. Pros - The proposed model leads to improvements in classification performance for small datasets and tasks with fine categories. Cons - The model appears to be overly complicated. Is just one gating function that includes the dependence on y (W_phi) really not sufficient ? It would be great to see empirical evidence that this complexity is useful. - The deterministic attention baseline seems to be using gating derived from x alone (without the second gating used in the proposed dropmax method which also regresses to y). It would be useful to have a clear comparison between dropmax and a deterministic version of dropmax. - The second gating function is essentially doing the same task as classification but with a unit-wise cross-entropy loss instead of a multiclass cross-entropy loss which the main classifier is using. The resulting model is a product of the two classifiers. It is plausible that the reported gains are a result of this ensembling of 2 classifiers, and not the dropout regularization. A determinstic baseline (as suggested above) would help resolve this question. Overall : The model is presented as a new variant of dropout derived using a somewhat complicated setup, when it can perhaps be more simply understood as ensembling a unit-wise logitic regression loss and the familiar softmax loss. It is not clear if the current presentation is the cleanest way to describe this model. However, the model is sound and leads to incremental improvements.